A pre-averaged pseudo nearest neighbor classifier

Li Dapeng lidp401@163.com
School of Software Engineering, Jinling Institute of Technology , Nanjing , China
Hong Tzung-Pei
Electronic publication date: 2024 Aug 13
Publication date: 2024
Volume: 10
Electronic Location ID: e2247
Received 2024 Feb 2; Accepted 2024 Jul 17
Copyright: © 2024 Li
Copyright year: 2024
Copyright holder: Li
License: This is an open access article distributed under the terms of the Creative Commons Attribution License, which permits unrestricted use, distribution, reproduction and adaptation in any medium and for any purpose provided that it is properly attributed. For attribution, the original author(s), title, publication source (PeerJ Computer Science) and either DOI or URL of the article must be cited.
License URL: https://creativecommons.org/licenses/by/4.0/

Keywords: Pre-averaged, Pseudo nearest neighbors, Small-size samples

Funding: The authors received no funding for this work.

==============================
The k-nearest neighbor algorithm is a powerful classification method. However, its classification performance will be affected in small-size samples with existing outliers. To address this issue, a pre-averaged pseudo nearest neighbor classifier (PAPNN) is proposed to improve classification performance. In the PAPNN rule, the pre-averaged categorical vectors are calculated by taking the average of any two points of the training sets in each class. Then, k-pseudo nearest neighbors are chosen from the preprocessed vectors of every class to determine the category of a query point. The pre-averaged vectors can reduce the negative impact of outliers to some degree. Extensive experiments are conducted on nineteen numerical real data sets and three high dimensional real data sets by comparing PAPNN to other twelve classification methods. The experimental results demonstrate that the proposed PAPNN rule is effective for classification tasks in the case of small-size samples with existing outliers.

Introduction

Supervised learning is an important field of machine learning. Random forest (RF) (Breiman, 2001), support vector machine (SVM) (Cortes & Vapnik, 1995), and k-nearest neighbor (kNN) (Cover & Hart, 1967) are common supervised learning methods. The kNN algorithm is a simple and effective machine learning method. It was first proposed for classification (Cover & Hart, 1967) in 1967. This algorithm assigns a test sample to the class represented by most of the k-nearest neighbors in the training set. In the kNN rule, the asymptotic classification performance can be achieved using the Bayes method under sufficient conditions (Li, Chen & Chen, 2008). It is one of the top 10 classification algorithms in the data mining field (Wu et al., 2008). Hence, kNN has been widely studied. Many kNN-based classification methods have been proposed (Li, Chen & Chen, 2008; Pan, Wang & Ku, 2017; Memis, Enginoǧlu & Erkan, 2022b; Zeng, Yang & Zhao, 2009a; Chai et al., 2021; Memis, Enginoǧlu & Erkan, 2022a; Memis, 2023; Keller, Gray & Givens, 1985; Mullick, Datta & Das, 2018; Memis, Enginoǧlu & Erkan, 2022c; Erkan, 2021; Kumbure, Luukka & Collan, 2018). Only one parameter, k, can be selected for kNN rule. If k is too small or large, the classifier is sensitive to noise points and outliers (Gou et al., 2019c). Hence, the classification accuracy of kNN is easily affected by outliers, especially in the case of small size training sets (Zhang, 2022).

To deal with the issue, some effective methods have been reported to reduce the negative impact of outliers. A local mean-based nearest neighbor classifier (Mitani & Hamamoto, 2006) (LMkNN) has been proposed. The categorical mean vectors in LMkNN rule can somewhat counteract the influence of outliers. The pattern of a test point is determined according to the weighted distance sum of k-nearest points chosen from each class in the pseudo nearest neighbor (Zeng, Yang & Zhao, 2009b) (PNN) rule, based on the distance weighted k-nearest neighbor (WkNN) rule (Dudani, 1976). The classification performance is better than kNN in small-size training sample case with outliers.

As an extension of both LMkNN and PNN, a local mean-based pseudo nearest neighbor classifier (LMPNN) was proposed (Gou et al., 2014). The category of a test point is predicted according to the weighted distance sum between the k-pseudo nearest neighbors and the test point in LMPNN classifier rule. The classification accuracy is higher than kNN, LMkNN, and PNN. Based on the ideas of PNN, LMkNN, and LMPNN, some derived methods have been reported (Pan, Wang & Ku, 2017; Gou et al., 2022, 2019b; Ma et al., 2022; Gou et al., 2019a; Pan et al., 2021; Gou et al., 2019c; Zhang et al., 2019; Gou et al., 2012; Gong et al., 2023; Xu et al., 2013) to improve classification performance.

Although several kNN-based classification methods have been proposed, how to further reduce the negative impact of outliers is still an open issue. Based on the above methods, pre-averaged pseudo-pre-averaged vectors from training samples are used. Then, the test sample is classified according to the distances between the first k-pseudo local nearest vectors and the test sample. Unlike the PNN method, the k-nearest neighbors are selected from the pre-averaged vectors in each class to determine the query point pattern in the PAPNN rule. The selected k-pseudo local nearest neighbors can further reduce the negative impact of outliers to some degree. The effectiveness and superiority of the proposed method are verified by experiments on numerical real sets and high dimensional real data sets.

The key work in this article is summarized as follows: 1. A pre-averaged pseudo nearest neighbor classifier is proposed. In this method, the categorical mean vectors are obtained by taking an average of two random points from training samples in each class. Then, the first k-pseudo local nearest neighbors from pre-averaged vectors of each class are selected in the PAPNN rule. The selected k-pseudo nearest neighbors are somewhat less sensitive to outliers. Hence, compared to conventional kNN, PAPNN can improve classification performance in the case of small-size samples with existing outliers. 2. Experiments are conducted to verify the effectiveness and superiority of PAPNN.

The remainder of this article is structured as follows. In the “Related Work” is briefly summarized. In the “Proposed PAPNN Method” is introduced. “Experiment” presents extensive experiments on numerical data sets and high dimensional data sets. Finally, a brief conclusion is drawn in “Conclusion”.

Related work

In this section, some related typical kNN-based classifiers are briefly reviewed. Portions of this text were previously published as part of a preprint (Li, 2024).

PNN classifier

PNN rule utilizes k-nearest neighbors from each class to determine a query point category. More class information can be captured compared to the traditional kNN. Hence, PNN can obtain better performance than kNN. The algorithm is described as follows: (1) Calculate the distance between the training sample yji and a test sample y from each class by euclidean distance:

(1) d(y,yji)=(y−yji)T(y−yji),

where yji represents the ith sample of class ωj, j=1,2,...,M, M represents the number of classes. Then, the first k-nearest distances d1j, d2j,…, dkj for class ωj can be obtained by sorting d(y,yji) in an ascending order.

(2) Assume that yjPNN denotes the pseudo neighbor from class ωj. The distance d(yjPNN,y) can be written as:

(2) d(yjPNN,y)=w1d1j+w2d2j+...+wkdkj,

where w1,w2,...,wk can be defined as wi=1/i according to PNN rule.

(3) The test sample y is classified into the class with the minimum distance of d(yjPNN,y) among all classes.

LMkNN classifier

As an extension of kNN, LMkNN (Mitani & Hamamoto, 2006) is a robust and simple nonparametric classifier. It is less sensitive to k than kNN. Point average method in LMkNN rule can reduce the influence of outliers in some degree, especially in the case of small-size training samples with existing outliers.

A query sample y∈RD in a D dimensional feature space is classified into class ωi by the following steps:

Step 1. Calculate the euclidean distances between the test sample y and training samples from each class ωi. Then, k-nearest neighbors y1i, y2i,…, yki in each class can be obtained, where i denotes a class number, according to the ascending order of the calculated distance.

Step 2. Calculate the local mean vector yi for every class:

(3) yi=1k∑j=1kyji.

Step 3. Calculate the distance between the local mean vector yi and the query point y for class ωi. Then, the pattern of the query sample y is finally classified into the class with the minimum distance between each categorical local mean vector and y among all classes.

LMPNN classifier

In LMPNN rule, multi local-mean based pseudo nearest neighbors in each class are utilized to determine a query sample pattern. Compared to PNN and LMkNN, LMPNN can obtain better classification accuracy rate in the case of small-size training samples with outliers. LMPNN rule is carried out as follows:

Step 1. Compute the distances between training samples in each class j and a query point y:

(4) d(y,yji)=(y−yji)T(y−yji),

where yji represents the ith training sample in class j. Then, the k-nearest neighbors ( y1j, y2j,…, ykj) of y for each class can be found by sorting the distances.

Step 2. Compute k local mean vectors for each class:

(5) xij¯=1i∑f=1iyfi,j=1,2,...,k,

where xij¯ denotes the ith local mean vector of class j.

Step 3. Let xjPNN¯ represents the categorical pseudo nearest neighbor of y from class j. The weighted distances sum between the local mean-based pseudo nearest neighbor xjPNN¯ and the unclasified point y among each class are computed as:

(6) d(y,xjPNN¯)=d(y,x1j¯)+(12d(y,x2j¯))+...+(1kd(y,xkj¯)).

Step 4. The pattern of the query point y is classified into the class w when the euclidean distance d(y,xwPNN¯) is minimum among all classes. Note that LMPNN, PNN, LMkNN is equivalent with k=1.

The proposed papnn classifier

In this section, a pre-averaged pseudo nearest neighbor classifier based on kNN is introduced. The purpose of PAPNN algorithm is to improve classification performance in the case of small-size training samples with existing outliers. Portions of this text were previously published as part of a preprint (Li, 2024).

The basic idea

In kNN rule, the classification accuracy is easily affected by outliers, especially in the case of small-size training points. The selection of the value k significantly influences classification performance. A small k may result in outliers being chosen as nearest neighbors, thereby compromising classification performance (Gou et al., 2019c). Conversely, in certain cases, the accuracy may be degraded with a large k value, as it incorporates numerous points from other classes among the k-nearest neighbors (Gou et al., 2019c).

The methods, such as LMkNN, PNN, and LPMNN, have been reported to improve classification performance in the case of small-size training samples with existing outliers. LMkNN can overcome the negative impact of the existing outliers to some degree. However, the same number of nearest neighbors for each class and the same weight coefficients for the nearest neighbors may affect classification effect (Gou et al., 2019c; Pan, Wang & Ku, 2017). In the PNN rule, larger weight coefficients are assigned to closer neighbors, which reduces the negative impact of outliers to some extent. Nevertheless, outliers can still compromise classification performance when the value of k is excessively large. LMPNN employs multiple pseudo local mean vectors derived from the k-nearest neighbors of each class to categorize a query point. Compared to LMkNN (Gou et al., 2014), LMPNN captures more class-specific information, thereby resulting in superior classification performance.

Based on the ideas of the aforesaid article, a pre-averaged based pseudo nearest neighbor classifier is introduced to reduce the adverse effects of outliers in the case of small-size training samples with existing outliers.

PAPNN rule

Let S=(yj,ci)j=1N be a training set with N training samples for M classes, where i=1,2,...,M, yj∈Rd, d is the feature dimension. Si=(yij,ci)j=1ni stands for the training sample set of class ci, where ni is the number of the training points in class ci. In the proposed PAPNN rule, the class label of an unclassified point y can be assigned by the following steps: 1. Calculate the pre-averaged vectors in each class by the following formula:

(7) xiq=(yij+yin)/2,j=0,1,...,(ni−1),n=(j+1),...,ni,q=1,2,...,ni(ni−1)/2,ni>1,

where xiq denotes the qth averaged vector of class ci. Note that xiq=yi1 with ni=1. 2. Compute the distances between the pre-averaged vectors in each class ci and y:

(8) d(y,xiq)=(y−xiq)T(y−xiq).

Then, the smallest k distances (d(y,xi1¯),d(y,xi2¯),...,d(y,xik¯)) can be obtained by sorting d(y,xiq) in an ascending order. 3. Let yiPAPNN denotes the categorical pseudo nearest neighbor of y from class ci. The weighted distance sum between the yiPAPNN and y among all classes are computed as:

(9) d(y,yiPAPNN)=∑j=1k1jd(y,xij¯).

4. Classify y as the class cj with the minimal distance:

(10) cj=argmind(y,yjPAPNN).

It should be noted that PAPNN, LMkNN, PNN, and LMPNN are equivalent with k=1. As mentioned above, the proposed PAPNN method is shown in Algorithm 1.

Algorithm 1 Outline of the proposed algorithm.

Require:	
   y: an unclassified point, k: the number of nearest neighbors.	
   Y=y1,y2,...,yn: n training points for m classes.	
   Yj=y1j,y2j,...,ynj: represents a training subset from the class cj with nj	
  training samples.	
 C = { c1, c2,…, cM}: the set of M classes.	
Ensure:	
 The class label of the unclassified point y.	
 Step 1: Calculate the pre-averaged vectors for each class with k>1.	
 for i = 1 to M do	
      q=0;	
   for j = 0 to (nj−1) do	
   for n = j+1 to nj do	
          xiq = (yij+yin)/2, q++;	
   end for	
  end for	
 end for	
 Note that xi1=yj1 with k=1.	
 Step 2: Calculate the distances between nj(nj−1)/2 mean vectors in each	
 class cj and y.	
 for q = 1 to (nj(nj−1)/2) do	
     d(y,xjq)=(y−xjq)T(y−xjq)	
 end for	
 Then, find first k pseudo local nearest points of class cj, denoted as yjPAPNN =	
  { xj1¯, xj2¯,…, xjk¯}.	
 Step 3: Compute the weighted distance sum between the first k pseudo nearest	
 neighbors and the query point y among each class.	
   for j = 1 to M do	
    for i = 1 to k do	
         d(y,yjPAPNN) += 1id(y,xji¯)	
    end for	
   end for	
  Step 4: The query sample y is classified into the category with the minimal	
  distance.	
   cjy = argmind(y,yjPAPNN).	

Experiments

To validate the classification performance of PAPNN rule, the extensive experiments on numerical real data sets and three high dimensional data sets are carried out to evaluate the performance of PAPNN method.

Data sets

In this subsection, the information of the selected data sets used in the experiments is presented.

The nineteen numerical real world data sets are taken from the UCI Machine Learning Repository (Bache & Lichman, 2013) and the KEEL Repository (Alcalá-Fdez et al., 2011), which are Vehicle, Balance, Blood, Bupa, Ionosphere, Pima-indians, Parkinsons, Hill, Haberman-survival, Musk-1, Sonar, Wine, Cardiotocography, QSAR, Band, Pima, Wine(keel), Mammographic and Steel, respectively. Among these nineteen real-world data sets, The numbers of total samples, attributes, classes of each data set, sources, and whether it is imbalanced are also listed in the first 19 rows of Table 1. From the first 19 rows Table 1, it can be seen that the data sets have different characteristics in numbers of attributes, samples, and classes. The numbers of all samples in the selected data sets are mostly small. These data sets can be used to verify the proposed method classification performance in the small training sample size cases.

Table 1 The data sets used in the experiments.

Data	Samples	Attributes	Classes	Source	Imbalance	
Vehicle	846	18	4	UCI	Yes	
Balance	625	4	3	UCI	Yes	
Blood	748	4	2	UCI	Yes	
Bupa	345	6	2	UCI	Yes	
Ionosphere	351	34	2	UCI	Yes	
Pima-Indians	768	8	2	UCI	Yes	
Parkinsons	195	22	2	UCI	Yes	
Hill	1,210	100	2	UCI	No	
Haberman-survival	306	3	2	UCI	Yes	
Musk-1	476	166	2	UCI	Yes	
Sonar	208	60	2	UCI	Yes	
Wine	178	13	3	UCI	Yes	
Cardiotocography	2,126	21	10	UCI	Yes	
QSAR	1,055	41	2	UCI	Yes	
Band	365	19	2	KEEL	Yes	
Pima	768	8	2	KEEL	Yes	
Wine(keel)	178	13	3	KEEL	Yes	
Mammographic	830	5	2	KEEL	Yes	
Steel	1,941	27	7	UCI	Yes	
AndrogenReceptor	1,687	1,024	2	UCI	Yes	
Semeion	1,593	256	2	UCI	No	
CNAE-9	1,080	857	9	UCI	Yes	

Experiments on the real data sets

Cross-validation is a common method to evaluate the performance of machine learning models. Among the various folds of cross-validation, five-fold and 10-fold cross-validation are commonly used (Erkan, 2021), (Kumbure, Luukka & Collan, 2018). Compared to five-fold cross-validation, 10-fold cross-validation can obtain more reliable performance results. Therefore, 10-fold cross validation is adopted in the experiments. Afterward, to ensure the reliability of performance results, 10 runs are carried out, and the average results are obtained for a specific value of the nearest neighbor k. The value of the nearest neighbor parameter k are varied from 1 to 20 with a step of 1. The classification results are determined by averaging the classification results of k=1,2,...,20. Experiments were performed by utilizing Python 3.7.3 and a computer with Intel(R) Core(TM) CPU i5-10210U @ 1.60GHz and 8 GB RAM.

Table 2 shows the comparison results between PAPNN and the other methods, i.e. kNN (Cover & Hart, 1967), WkNN (Dudani, 1976), PNN (Zeng, Yang & Zhao, 2009b), LMkNN (Mitani & Hamamoto, 2006), LMPNN (Gou et al., 2014), fuzzy kNN (FkNN) (Keller, Gray & Givens, 1985), eigenvalue classification method (EigenClass) (Erkan, 2021), generalized mean distance-based k-nearest neighbor classifier (GMDkNN) (Gou et al., 2019a), fuzzy k-nearest neighbor classifier based on the Bonferroni mean (BMFkNN) (Kumbure, Luukka & Collan, 2018), fuzzy parameterized fuzzy soft k-nearest neighbor classifier (FPFS-kNN) (Memis, Enginoǧlu & Erkan, 2022b), SVM (Cortes & Vapnik, 1995), and random forests (RF) (Breiman, 2001). The settings of the compared algorithms are summarised in Table 3.

Table 2 Comparative results for the data sets.

Data	Methods	Accuracy ± SD	Precision ± SD	Recall ± SD	F1-score ± SD	
Vehicle	kNN	0.7028 ± 0.0020	0.7016 ± 0.0022	0.7028 ± 0.0020	0.6898 ± 0.0012	
	WkNN	0.7146 ± 0.0016	0.7164 ± 0.0017	0.7146 ± 0.0016	0.7080 ± 0.0011	
	PNN	0.7152 ± 0.0012	0.7150 ± 0.0016	0.7152 ± 0.0012	0.7042 ± 0.0015	
	LMkNN	0.7324 ± 0.0015	0.7436 ± 0.0014	0.7324 ± 0.0015	0.7313 ± 0.0015	
	LMPNN	0.7324 ± 0.0015	0.7436 ± 0.0014	0.7324 ± 0.0015	0.7313 ± 0.0015	
	PAPNN	0.7777 ± 0.0007	0.7851 ± 0.0006	0.7777 ± 0.0007	0.7761 ± 0.0009	
	FkNN	0.7103 ± 0.0017	0.7106 ± 0.0021	0.7103 ± 0.0017	0.6990 ± 0.0019	
	EigenClass	0.6782 ± 0.0016	0.6744 ± 0.0016	0.6782 ± 0.0016	0.6660 ± 0.0010	
	GMDkNN	0.7315 ± 0.0012	0.7397 ± 0.0011	0.7315 ± 0.0012	0.7280 ± 0.0012	
	BMFkNN	0.7312 ± 0.0014	0.7426 ± 0.0014	0.7312 ± 0.0014	0.7304 ± 0.0009	
	FPFS-kNN	0.6585 ± 0.0022	0.6498 ± 0.0029	0.6585 ± 0.0022	0.6397 ± 0.0012	
	SVM	0.7245 ± 0.0008	0.7210 ± 0.0013	0.7245 ± 0.0008	0.7117 ± 0.0011	
	RF	0.7517 ± 0.0002	0.7539 ± 0.0003	0.7517 ± 0.0002	0.7457 ± 0.0002	
Balance	kNN	0.8705 ± 0.0016	0.8282 ± 0.0031	0.8705 ± 0.0016	0.8446 ± 0.0003	
	WkNN	0.8278 ± 0.0022	0.8034 ± 0.0038	0.8278 ± 0.0022	0.8144 ± 0.0008	
	PNN	0.8932 ± 0.0010	0.8330 ± 0.0029	0.8932 ± 0.0010	0.8594 ± 0.0002	
	LMkNN	0.9000 ± 0.0009	0.8903 ± 0.0019	0.9000 ± 0.0009	0.8853 ± 0.0074	
	LMPNN	0.9000 ± 0.0009	0.8903 ± 0.0019	0.9000 ± 0.0009	0.8853 ± 0.0074	
	PAPNN	0.9456 ± 0.0006	0.9469 ± 0.0006	0.9456 ± 0.0006	0.9422 ± 0.0043	
	FkNN	0.8721 ± 0.0017	0.8232 ± 0.0037	0.8721 ± 0.0017	0.8458 ± 0.0001	
	EigenClass	0.8647 ± 0.0014	0.8091 ± 0.0031	0.8647 ± 0.0014	0.8343 ± 0.0002	
	GMDkNN	0.8737 ± 0.0007	0.8348 ± 0.0028	0.8737 ± 0.0007	0.8503 ± 0.0006	
	BMFkNN	0.8399 ± 0.0010	0.8482 ± 0.0022	0.8399 ± 0.0010	0.8408 ± 0.0052	
	FPFS-kNN	0.8890 ± 0.0010	0.8283 ± 0.0028	0.8890 ± 0.0010	0.8565 ± 0.0003	
	SVM	0.8849 ± 0.0015	0.8195 ± 0.0033	0.8849 ± 0.0015	0.8493 ± 0.0023	
	RF	0.8193 ± 0.0014	0.8281 ± 0.0031	0.8193 ± 0.0014	0.8225 ± 0.0020	
Blood	kNN	0.7634 ± 0.0022	0.7450 ± 0.0037	0.7634 ± 0.0022	0.7413 ± 0.0040	
	WkNN	0.7414 ± 0.0018	0.7229 ± 0.0034	0.7414 ± 0.0018	0.7258 ± 0.0031	
	PNN	0.7567 ± 0.0018	0.7301 ± 0.0033	0.7567 ± 0.0018	0.7331 ± 0.0034	
	LMkNN	0.7609 ± 0.0015	0.7482 ± 0.0024	0.7609 ± 0.0015	0.7487 ± 0.0037	
	LMPNN	0.7609 ± 0.0015	0.7482 ± 0.0024	0.7609 ± 0.0015	0.7487 ± 0.0037	
	PAPNN	0.7837 ± 0.0021	0.7614 ± 0.0040	0.7837 ± 0.0021	0.7507 ± 0.0014	
	FkNN	0.7085 ± 0.0023	0.7070 ± 0.0027	0.7085 ± 0.0023	0.7029 ± 0.0037	
	EigenClass	0.7762 ± 0.0019	0.7528 ± 0.0032	0.7762 ± 0.0019	0.7550 ± 0.0022	
	GMDkNN	0.7497 ± 0.0017	0.7294 ± 0.0034	0.7497 ± 0.0017	0.7343 ± 0.0026	
	BMFkNN	0.6509 ± 0.0020	0.7027 ± 0.0034	0.6509 ± 0.0020	0.6674 ± 0.0047	
	FPFS-kNN	0.7781 ± 0.0014	0.7571 ± 0.0026	0.7781 ± 0.0014	0.7573 ± 0.0031	
	SVM	0.7620 ± 0.0013	0.5820 ± 0.0031	0.7620 ± 0.0013	0.6595 ± 0.0024	
	RF	0.7459 ± 0.0014	0.7235 ± 0.0032	0.7459 ± 0.0014	0.7283 ± 0.0022	
Bupa	kNN	0.6411 ± 0.0055	0.6530 ± 0.0067	0.6411 ± 0.0055	0.6360 ± 0.0011	
	WkNN	0.6351 ± 0.0077	0.6501 ± 0.0102	0.6351 ± 0.0077	0.6312 ± 0.0069	
	PNN	0.6479 ± 0.0066	0.6610 ± 0.0075	0.6479 ± 0.0066	0.6442 ± 0.0048	
	LMkNN	0.6585 ± 0.0048	0.6683 ± 0.0060	0.6585 ± 0.0048	0.6454 ± 0.0041	
	LMPNN	0.6585 ± 0.0048	0.6683 ± 0.0060	0.6585 ± 0.0048	0.6454 ± 0.0041	
	PAPNN	0.6825 ± 0.0059	0.6858 ± 0.0064	0.6825 ± 0.0059	0.6764 ± 0.0072	
	FkNN	0.6451 ± 0.0048	0.6618 ± 0.0061	0.6451 ± 0.0048	0.6389 ± 0.0019	
	EigenClass	0.688 ± 0.0054	0.7061 ± 0.0056	0.688 ± 0.0054	0.6752 ± 0.0035	
	GMDkNN	0.6349 ± 0.0058	0.6458 ± 0.0063	0.6349 ± 0.0058	0.6307 ± 0.0071	
	BMFkNN	0.6190 ± 0.0075	0.6276 ± 0.0079	0.6190 ± 0.0075	0.6160 ± 0.0106	
	FPFS-kNN	0.6030 ± 0.0061	0.6130 ± 0.0070	0.6030 ± 0.0061	0.5835 ± 0.0034	
	SVM	0.5824 ± 0.0057	0.3709 ± 0.0254	0.5824 ± 0.0057	0.4339 ± 0.0097	
	RF	0.7481 ± 0.0025	0.7628 ± 0.0024	0.7481 ± 0.0025	0.7419 ± 0.0035	
Ionosphere	kNN	0.8426 ± 0.0036	0.8652 ± 0.0028	0.8426 ± 0.0036	0.8304 ± 0.0076	
	WkNN	0.8487 ± 0.0032	0.8683 ± 0.0030	0.8487 ± 0.0032	0.8368 ± 0.0097	
	PNN	0.8562 ± 0.0025	0.8738 ± 0.0024	0.8562 ± 0.0025	0.8448 ± 0.0072	
	LMkNN	0.8958 ± 0.0023	0.9055 ± 0.0020	0.8958 ± 0.0023	0.8904 ± 0.0050	
	LMPNN	0.8958 ± 0.0023	0.9055 ± 0.0020	0.8958 ± 0.0023	0.8904 ± 0.0050	
	PAPNN	0.9215 ± 0.0018	0.9289 ± 0.0016	0.9215 ± 0.0018	0.9190 ± 0.0048	
	FkNN	0.8443 ± 0.0034	0.8658 ± 0.0028	0.8443 ± 0.0034	0.8322 ± 0.0079	
	EigenClass	0.9306 ± 0.0040	0.9316 ± 0.0045	0.9306 ± 0.0040	0.9300 ± 0.0124	
	GMDkNN	0.8835 ± 0.0021	0.8936 ± 0.0024	0.8835 ± 0.0021	0.8778 ± 0.0058	
	BMFkNN	0.8839 ± 0.0025	0.8923 ± 0.0033	0.8839 ± 0.0025	0.8773 ± 0.0048	
	FPFS-kNN	0.6415 ± 0.0119	0.4234 ± 0.0157	0.6415 ± 0.0119	0.5073 ± 0.0152	
	SVM	0.8830 ± 0.0022	0.8902 ± 0.0020	0.8830 ± 0.0022	0.8775 ± 0.0025	
	RF	0.9315 ± 0.0008	0.9345 ± 0.0007	0.9315 ± 0.0008	0.9313 ± 0.0008	
Pima-Indians	kNN	0.7375 ± 0.0019	0.7322 ± 0.0022	0.7375 ± 0.0019	0.7293 ± 0.0021	
	WkNN	0.7327 ± 0.0032	0.7274 ± 0.0038	0.7327 ± 0.0032	0.7260 ± 0.0034	
	PNN	0.7321 ± 0.0031	0.7276 ± 0.0039	0.7321 ± 0.0031	0.7246 ± 0.0028	
	LMkNN	0.7417 ± 0.0020	0.7437 ± 0.0019	0.7417 ± 0.0020	0.7405 ± 0.0019	
	LMPNN	0.7417 ± 0.0020	0.7437 ± 0.0019	0.7417 ± 0.0020	0.7405 ± 0.0019	
	PAPNN	0.7473 ± 0.0033	0.7433 ± 0.0035	0.7473 ± 0.0033	0.7412 ± 0.0038	
	FkNN	0.7359 ± 0.0019	0.7313 ± 0.0021	0.7359 ± 0.0019	0.7290 ± 0.0021	
	EigenClass	0.7031 ± 0.0028	0.6957 ± 0.0044	0.7031 ± 0.0028	0.6721 ± 0.0018	
	GMDkNN	0.7256 ± 0.0034	0.7224 ± 0.0038	0.7256 ± 0.0034	0.7218 ± 0.0029	
	BMFkNN	0.7078 ± 0.0026	0.7041 ± 0.0029	0.7078 ± 0.0026	0.7031 ± 0.0025	
	FPFS-kNN	0.7446 ± 0.0021	0.7406 ± 0.0025	0.7446 ± 0.0021	0.7364 ± 0.0029	
	SVM	0.7708 ± 0.0018	0.7706 ± 0.0021	0.7708 ± 0.0018	0.7593 ± 0.0018	
	RF	0.7747 ± 0.0025	0.7731 ± 0.0027	0.7747 ± 0.0025	0.7696 ± 0.0027	
Parkinsons	kNN	0.8877 ± 0.0036	0.8911 ± 0.0054	0.8877 ± 0.0036	0.8743 ± 0.0117	
	WkNN	0.9355 ± 0.0011	0.9414 ± 0.0010	0.9355 ± 0.0011	0.9338 ± 0.0047	
	PNN	0.9268 ± 0.0015	0.9341 ± 0.0012	0.9268 ± 0.0015	0.9244 ± 0.0097	
	LMkNN	0.8974 ± 0.0037	0.9077 ± 0.0029	0.8974 ± 0.0037	0.8876 ± 0.0126	
	LMPNN	0.8974 ± 0.0037	0.9077 ± 0.0029	0.8974 ± 0.0037	0.8876 ± 0.0126	
	PAPNN	0.9356 ± 0.0026	0.9400 ± 0.0024	0.9356 ± 0.0026	0.9344 ± 0.0061	
	FkNN	0.9340 ± 0.0017	0.9412 ± 0.0014	0.9340 ± 0.0017	0.9317 ± 0.0095	
	EigenClass	0.8703 ± 0.0031	0.8782 ± 0.0031	0.8703 ± 0.0031	0.8639 ± 0.0102	
	GMDkNN	0.9526 ± 0.0011	0.9583 ± 0.0009	0.9526 ± 0.0011	0.9515 ± 0.0048	
	BMFkNN	0.9412 ± 0.0021	0.9487 ± 0.0016	0.9412 ± 0.0021	0.9416 ± 0.0063	
	FPFS-kNN	0.8969 ± 0.0029	0.9096 ± 0.0023	0.8969 ± 0.0029	0.8848 ± 0.0121	
	SVM	0.8715 ± 0.0054	0.8874 ± 0.0042	0.8715 ± 0.0054	0.8505 ± 0.0086	
	RF	0.9026 ± 0.0027	0.9178 ± 0.0014	0.9026 ± 0.0027	0.8962 ± 0.0034	
Hill	kNN	0.5305 ± 0.0020	0.5363 ± 0.0020	0.5305 ± 0.0020	0.5302 ± 0.0025	
	WkNN	0.5748 ± 0.0013	0.5800 ± 0.0015	0.5748 ± 0.0013	0.5746 ± 0.0019	
	PNN	0.5642 ± 0.0013	0.5701 ± 0.0015	0.5642 ± 0.0013	0.5641 ± 0.0016	
	LMkNN	0.6074 ± 0.0037	0.6113 ± 0.0037	0.6074 ± 0.0037	0.6074 ± 0.0041	
	LMPNN	0.6074 ± 0.0037	0.6113 ± 0.0037	0.6074 ± 0.0037	0.6074 ± 0.0041	
	PAPNN	0.9447 ± 0.0005	0.9457 ± 0.0004	0.9447 ± 0.0005	0.9448 ± 0.0005	
	FkNN	0.5453 ± 0.0019	0.5506 ± 0.0020	0.5453 ± 0.0019	0.5447 ± 0.0016	
	EigenClass	0.5293 ± 0.0018	0.5354 ± 0.0018	0.5293 ± 0.0018	0.5289 ± 0.0015	
	GMDkNN	0.6258 ± 0.0022	0.6296 ± 0.0024	0.6258 ± 0.0022	0.6257 ± 0.0024	
	BMFkNN	0.6215 ± 0.0030	0.6240 ± 0.0032	0.6215 ± 0.0030	0.6209 ± 0.0034	
	FPFS-kNN	0.4793 ± 0.0017	0.4858 ± 0.0016	0.4793 ± 0.0017	0.4785 ± 0.0010	
	SVM	0.4933 ± 0.0015	0.6581 ± 0.0218	0.4933 ± 0.0015	0.3599 ± 0.0026	
	RF	0.5569 ± 0.0030	0.5607 ± 0.0029	0.5569 ± 0.0030	0.5574 ± 0.0029	
Haberman-survival	kNN	0.7184 ± 0.0059	0.6894 ± 0.0108	0.7184 ± 0.0059	0.6737 ± 0.0044	
	WkNN	0.6886 ± 0.0026	0.6601 ± 0.0058	0.6886 ± 0.0026	0.6563 ± 0.0032	
	PNN	0.6949 ± 0.0041	0.6450 ± 0.0068	0.6949 ± 0.0041	0.6481 ± 0.0014	
	LMkNN	0.6973 ± 0.0058	0.7028 ± 0.0082	0.6973 ± 0.0058	0.6823 ± 0.0092	
	LMPNN	0.6973 ± 0.0058	0.7028 ± 0.0082	0.6973 ± 0.0058	0.6824 ± 0.0092	
	PAPNN	0.7134 ± 0.0081	0.6505 ± 0.0291	0.7134 ± 0.0081	0.6553 ± 0.0039	
	FkNN	0.686 ± 0.0024	0.6652 ± 0.0044	0.686 ± 0.0024	0.6594 ± 0.0032	
	EigenClass	0.7014 ± 0.0078	0.6525 ± 0.0187	0.7014 ± 0.0078	0.6456 ± 0.0024	
	GMDkNN	0.6686 ± 0.0041	0.6670 ± 0.0075	0.6686 ± 0.0041	0.6501 ± 0.0046	
	BMFkNN	0.6381 ± 0.0049	0.6615 ± 0.0080	0.6381 ± 0.0049	0.6425 ± 0.0077	
	FPFS-kNN	0.7136 ± 0.0068	0.7184 ± 0.0105	0.7136 ± 0.0068	0.6840 ± 0.0061	
	SVM	0.7297 ± 0.0071	0.5506 ± 0.0192	0.7297 ± 0.0071	0.6246 ± 0.0144	
	RF	0.6801 ± 0.0054	0.6784 ± 0.0094	0.6801 ± 0.0054	0.6632 ± 0.0071	
Musk-1	kNN	0.8072 ± 0.0032	0.8359 ± 0.0020	0.8072 ± 0.0032	0.8078 ± 0.0011	
	WkNN	0.8477 ± 0.0031	0.8660 ± 0.0020	0.8477 ± 0.0031	0.8489 ± 0.0022	
	PNN	0.8552 ± 0.0027	0.8738 ± 0.0018	0.8552 ± 0.0027	0.8562 ± 0.0024	
	LMkNN	0.8725 ± 0.0014	0.8800 ± 0.0012	0.8725 ± 0.0014	0.8723 ± 0.0008	
	LMPNN	0.8725 ± 0.0014	0.8800 ± 0.0012	0.8725 ± 0.0014	0.8723 ± 0.0008	
	PAPNN	0.9047 ± 0.0027	0.9098 ± 0.0022	0.9047 ± 0.0027	0.9053 ± 0.0026	
	FkNN	0.8323 ± 0.0036	0.8580 ± 0.0023	0.8323 ± 0.0036	0.8331 ± 0.0027	
	EigenClass	0.7089 ± 0.0024	0.7798 ± 0.0025	0.7089 ± 0.0024	0.7023 ± 0.0016	
	GMDkNN	0.8844 ± 0.0025	0.8906 ± 0.0020	0.8844 ± 0.0025	0.8852 ± 0.0020	
	BMFkNN	0.8797 ± 0.0026	0.8890 ± 0.0019	0.8797 ± 0.0026	0.8805 ± 0.0024	
	FPFS-kNN	0.8181 ± 0.0024	0.8255 ± 0.0021	0.8181 ± 0.0024	0.8187 ± 0.0010	
	SVM	0.8339 ± 0.0008	0.8374 ± 0.0008	0.8339 ± 0.0008	0.8338 ± 0.0008	
	RF	0.8908 ± 0.0015	0.8930 ± 0.0016	0.8908 ± 0.0015	0.8905 ± 0.0015	
Sonar	kNN	0.7618 ± 0.0079	0.7783 ± 0.0082	0.7618 ± 0.0079	0.7558 ± 0.0184	
	WkNN	0.8542 ± 0.0059	0.8672 ± 0.0045	0.8542 ± 0.0059	0.8533 ± 0.0039	
	PNN	0.8606 ± 0.0038	0.8761 ± 0.0029	0.8606 ± 0.0038	0.8589 ± 0.0036	
	LMkNN	0.8440 ± 0.0061	0.8625 ± 0.0051	0.8440 ± 0.0061	0.8437 ± 0.0062	
	LMPNN	0.8440 ± 0.0061	0.8625 ± 0.0051	0.8440 ± 0.0061	0.8437 ± 0.0062	
	PAPNN	0.8964 ± 0.0020	0.9040 ± 0.0019	0.8964 ± 0.0020	0.8965 ± 0.0022	
	FkNN	0.8155 ± 0.0051	0.8341 ± 0.0053	0.8155 ± 0.0051	0.8112 ± 0.0105	
	EigenClass	0.7524 ± 0.0084	0.8104 ± 0.0058	0.7524 ± 0.0084	0.7459 ± 0.0053	
	GMDkNN	0.8826 ± 0.0041	0.8937 ± 0.0035	0.8826 ± 0.0041	0.8828 ± 0.0025	
	BMFkNN	0.8773 ± 0.0029	0.8880 ± 0.0025	0.8773 ± 0.0029	0.8775 ± 0.0042	
	FPFS-kNN	0.8391 ± 0.0043	0.8608 ± 0.0036	0.8391 ± 0.0043	0.8366 ± 0.0049	
	SVM	0.7640 ± 0.0058	0.7912 ± 0.0060	0.7640 ± 0.0058	0.7631 ± 0.0057	
	RF	0.8364 ± 0.0020	0.8549 ± 0.0016	0.8364 ± 0.0020	0.8350 ± 0.0020	
Wine	kNN	0.9629 ± 0.0040	0.9731 ± 0.0023	0.9629 ± 0.0040	0.9630 ± 0.0011	
	WkNN	0.9588 ± 0.0042	0.9710 ± 0.0020	0.9588 ± 0.0042	0.9587 ± 0.0019	
	PNN	0.9644 ± 0.0036	0.9751 ± 0.0018	0.9644 ± 0.0036	0.9645 ± 0.0011	
	LMkNN	0.9767 ± 0.0018	0.9815 ± 0.0012	0.9767 ± 0.0018	0.9769 ± 0.0005	
	LMPNN	0.9767 ± 0.0018	0.9815 ± 0.0012	0.9767 ± 0.0018	0.9769 ± 0.0005	
	PAPNN	0.9844 ± 0.0009	0.9879 ± 0.0005	0.9844 ± 0.0009	0.9846 ± 0.0005	
	FkNN	0.9651 ± 0.0034	0.9756 ± 0.0017	0.9651 ± 0.0034	0.9654 ± 0.0011	
	EigenClass	0.9711 ± 0.0011	0.9815 ± 0.0004	0.9711 ± 0.0011	0.9733 ± 0.0007	
	GMDkNN	0.9733 ± 0.0025	0.9803 ± 0.0013	0.9733 ± 0.0025	0.9733 ± 0.0005	
	BMFkNN	0.9643 ± 0.0025	0.9705 ± 0.0018	0.9643 ± 0.0025	0.9643 ± 0.0021	
	FPFS-kNN	0.9521 ± 0.0024	0.9643 ± 0.0014	0.9521 ± 0.0024	0.9527 ± 0.0014	
	SVM	0.9833 ± 0.0006	0.9882 ± 0.0003	0.9833 ± 0.0006	0.9843 ± 0.0005	
	RF	0.9777 ± 0.0013	0.9813 ± 0.0011	0.9777 ± 0.0013	0.9785 ± 0.0013	
Cardiotocography	kNN	0.7511 ± 0.0005	0.7528 ± 0.0006	0.7511 ± 0.0005	0.7384 ± 0.0007	
	WkNN	0.7817 ± 0.0005	0.7833 ± 0.0006	0.7817 ± 0.0005	0.7756 ± 0.0014	
	PNN	0.7913 ± 0.0004	0.7956 ± 0.0006	0.7913 ± 0.0004	0.7817 ± 0.0010	
	LMkNN	0.7890 ± 0.0004	0.7969 ± 0.0006	0.7890 ± 0.0004	0.7822 ± 0.0012	
	LMPNN	0.7890 ± 0.0004	0.7969 ± 0.0006	0.7890 ± 0.0004	0.7822 ± 0.0012	
	PAPNN	0.8049 ± 0.0002	0.8143 ± 0.0002	0.8049 ± 0.0002	0.7987 ± 0.0011	
	FkNN	0.7860 ± 0.0003	0.7905 ± 0.0006	0.7860 ± 0.0003	0.7782 ± 0.0013	
	EigenClass	0.7285 ± 0.0005	0.7395 ± 0.0004	0.7285 ± 0.0005	0.7153 ± 0.0007	
	GMDkNN	0.8057 ± 0.0004	0.8102 ± 0.0005	0.8057 ± 0.0004	0.8015 ± 0.0012	
	BMFkNN	0.7883 ± 0.0005	0.7945 ± 0.0007	0.7883 ± 0.0005	0.7862 ± 0.0006	
	FPFS-kNN	0.7996 ± 0.0004	0.8010 ± 0.0007	0.7996 ± 0.0004	0.7860 ± 0.0013	
	SVM	0.7840 ± 0.0006	0.7742 ± 0.0007	0.7840 ± 0.0006	0.7637 ± 0.0006	
	RF	0.8763 ± 0.0002	0.8818 ± 0.0003	0.8763 ± 0.0002	0.8737 ± 0.0003	
QSAR	kNN	0.8522 ± 0.0011	0.8572 ± 0.0013	0.8522 ± 0.0011	0.8531 ± 0.0019	
	WkNN	0.8628 ± 0.0009	0.8668 ± 0.0009	0.8628 ± 0.0009	0.8635 ± 0.0010	
	PNN	0.8647 ± 0.0009	0.8678 ± 0.0009	0.8647 ± 0.0009	0.8653 ± 0.0016	
	LMkNN	0.8610 ± 0.0007	0.8619 ± 0.0007	0.8610 ± 0.0007	0.8597 ± 0.0004	
	LMPNN	0.8610 ± 0.0007	0.8619 ± 0.0007	0.8610 ± 0.0007	0.8597 ± 0.0004	
	PAPNN	0.8722 ± 0.0009	0.8734 ± 0.0010	0.8722 ± 0.0009	0.8708 ± 0.0009	
	FkNN	0.8549 ± 0.0009	0.8591 ± 0.0010	0.8549 ± 0.0009	0.8557 ± 0.0019	
	EigenClass	0.8072 ± 0.0011	0.8241 ± 0.0015	0.8072 ± 0.0011	0.8107 ± 0.0027	
	GMDkNN	0.8637 ± 0.0007	0.8650 ± 0.0008	0.8637 ± 0.0007	0.8635 ± 0.0013	
	BMFkNN	0.8269 ± 0.0008	0.8302 ± 0.0008	0.8269 ± 0.0008	0.8271 ± 0.0019	
	FPFS-kNN	0.8528 ± 0.0008	0.8559 ± 0.0009	0.8528 ± 0.0008	0.8526 ± 0.0008	
	SVM	0.8531 ± 0.0005	0.8557 ± 0.0005	0.8531 ± 0.0005	0.8508 ± 0.0006	
	RF	0.8702 ± 0.0007	0.8711 ± 0.0008	0.8702 ± 0.0007	0.8680 ± 0.0008	
Band	kNN	0.6792 ± 0.0055	0.6874 ± 0.0052	0.6792 ± 0.0055	0.6632 ± 0.0037	
	WkNN	0.7365 ± 0.0047	0.7548 ± 0.0051	0.7365 ± 0.0047	0.7313 ± 0.0032	
	PNN	0.7408 ± 0.0044	0.7595 ± 0.0038	0.7408 ± 0.0044	0.7336 ± 0.0039	
	LMkNN	0.7083 ± 0.0055	0.7220 ± 0.0052	0.7083 ± 0.0055	0.6980 ± 0.0031	
	LMPNN	0.7083 ± 0.0055	0.7220 ± 0.0052	0.7083 ± 0.0055	0.6980 ± 0.0031	
	PAPNN	0.7426 ± 0.0029	0.7654 ± 0.0025	0.7426 ± 0.0029	0.7289 ± 0.0038	
	FkNN	0.7059 ± 0.0055	0.7192 ± 0.0049	0.7059 ± 0.0055	0.6919 ± 0.0018	
	EigenClass	0.6806 ± 0.0093	0.6883 ± 0.0112	0.6806 ± 0.0093	0.6449 ± 0.0019	
	GMDkNN	0.7416 ± 0.0050	0.7626 ± 0.0048	0.7416 ± 0.0050	0.7408 ± 0.0055	
	BMFkNN	0.7078 ± 0.0071	0.7235 ± 0.0066	0.7078 ± 0.0071	0.7036 ± 0.0089	
	FPFS-kNN	0.6711 ± 0.0066	0.6786 ± 0.0055	0.6711 ± 0.0066	0.6546 ± 0.0050	
	SVM	0.6907 ± 0.0062	0.7092 ± 0.0076	0.6907 ± 0.0062	0.6488 ± 0.0083	
	RF	0.7261 ± 0.0071	0.7426 ± 0.0081	0.7261 ± 0.0071	0.7208 ± 0.0057	
Pima	kNN	0.7397 ± 0.0014	0.7379 ± 0.0014	0.7397 ± 0.0014	0.7327 ± 0.0014	
	WkNN	0.7311 ± 0.0013	0.7293 ± 0.0011	0.7311 ± 0.0013	0.7247 ± 0.0013	
	PNN	0.7336 ± 0.0012	0.7323 ± 0.0011	0.7336 ± 0.0012	0.7266 ± 0.0014	
	LMkNN	0.7414 ± 0.0018	0.7479 ± 0.0016	0.7414 ± 0.0018	0.7412 ± 0.0035	
	LMPNN	0.7414 ± 0.0018	0.7479 ± 0.0016	0.7414 ± 0.0018	0.7412 ± 0.0035	
	PAPNN	0.7551 ± 0.0013	0.7568 ± 0.0009	0.7551 ± 0.0013	0.7480 ± 0.0021	
	FkNN	0.7388 ± 0.0013	0.7381 ± 0.0012	0.7388 ± 0.0013	0.7332 ± 0.0013	
	EigenClass	0.6994 ± 0.0024	0.6934 ± 0.0020	0.6994 ± 0.0024	0.6652 ± 0.0016	
	GMDkNN	0.7264 ± 0.0016	0.7288 ± 0.0013	0.7264 ± 0.0016	0.7236 ± 0.0021	
	BMFkNN	0.7143 ± 0.0014	0.7168 ± 0.0013	0.7143 ± 0.0014	0.7100 ± 0.0024	
	FPFS-kNN	0.7410 ± 0.0012	0.7400 ± 0.0012	0.7410 ± 0.0012	0.7342 ± 0.0029	
	SVM	0.7681 ± 0.0013	0.7675 ± 0.0016	0.7681 ± 0.0013	0.7577 ± 0.0016	
	RF	0.7655 ± 0.0017	0.7629 ± 0.0018	0.7655 ± 0.0017	0.7594 ± 0.0018	
Wine (keel)	kNN	0.9660 ± 0.0018	0.9730 ± 0.0011	0.9660 ± 0.0018	0.9664 ± 0.0009	
	WkNN	0.9627 ± 0.0019	0.9706 ± 0.0011	0.9627 ± 0.0019	0.9627 ± 0.0014	
	PNN	0.9680 ± 0.0014	0.9745 ± 0.0009	0.9680 ± 0.0014	0.9680 ± 0.0011	
	LMkNN	0.9711 ± 0.0008	0.9769 ± 0.0005	0.9711 ± 0.0008	0.9715 ± 0.0003	
	LMPNN	0.9711 ± 0.0008	0.9769 ± 0.0057	0.9711 ± 0.0008	0.9715 ± 0.0003	
	PAPNN	0.9895 ± 0.0007	0.9879 ± 0.0004	0.9895 ± 0.0007	0.9843 ± 0.0004	
	FkNN	0.9680 ± 0.0016	0.9746 ± 0.0010	0.9680 ± 0.0016	0.9684 ± 0.0005	
	EigenClass	0.9695 ± 0.0014	0.9755 ± 0.0009	0.9695 ± 0.0014	0.9698 ± 0.0008	
	GMDkNN	0.9710 ± 0.0011	0.9770 ± 0.0007	0.9710 ± 0.0011	0.9716 ± 0.0004	
	BMFkNN	0.9679 ± 0.0014	0.9744 ± 0.0009	0.9679 ± 0.0014	0.9682 ± 0.0024	
	FPFS-kNN	0.9567 ± 0.0035	0.9667 ± 0.0019	0.9567 ± 0.0035	0.9566 ± 0.0021	
	SVM	0.9830 ± 0.0006	0.9872 ± 0.0003	0.9830 ± 0.0006	0.9836 ± 0.0006	
	RF	0.9830 ± 0.0012	0.9874 ± 0.0007	0.9830 ± 0.0012	0.9833 ± 0.0012	
Steel	kNN	0.7069 ± 0.0010	0.7140 ± 0.0010	0.7069 ± 0.0010	0.7044 ± 0.0019	
	WkNN	0.7202 ± 0.0008	0.7245 ± 0.0008	0.7202 ± 0.0008	0.7184 ± 0.0013	
	PNN	0.7209 ± 0.0010	0.7274 ± 0.0010	0.7209 ± 0.0010	0.7191 ± 0.0016	
	LMkNN	0.7144 ± 0.0008	0.7272 ± 0.0008	0.7144 ± 0.00087	0.71205 ± 0.0038	
	LMPNN	0.7144 ± 0.0008	0.7272 ± 0.0008	0.7144 ± 0.0008	0.7120 ± 0.0038	
	PAPNN	0.7405 ± 0.0008	0.7467 ± 0.0007	0.7405 ± 0.0008	0.7382 ± 0.0024	
	FkNN	0.7227 ± 0.0010	0.7279 ± 0.0010	0.7227 ± 0.0010	0.7205 ± 0.0017	
	EigenClass	0.6982 ± 0.0011	0.7140 ± 0.0011	0.6982 ± 0.0011	0.6904 ± 0.0020	
	GMDkNN	0.7318 ± 0.0008	0.7374 ± 0.0008	0.7318 ± 0.0008	0.7305 ± 0.0017	
	BMFkNN	0.7144 ± 0.0009	0.7208 ± 0.0010	0.7144 ± 0.0009	0.7136 ± 0.0025	
	FPFS-kNN	0.6972 ± 0.0009	0.7060 ± 0.0008	0.6972 ± 0.0009	0.6904 ± 0.0031	
	SVM	0.6939 ± 0.0008	0.6870 ± 0.0010	0.6939 ± 0.0008	0.6835 ± 0.0009	
	RF	0.7815 ± 0.0003	0.7894 ± 0.0002	0.7815 ± 0.0003	0.7807 ± 0.0002	
Mammographic	kNN	0.7965 ± 0.0012	0.8027 ± 0.0011	0.7965 ± 0.0012	0.7960 ± 0.0018	
	WkNN	0.7805 ± 0.0009	0.7859 ± 0.0010	0.7805 ± 0.0009	0.7802 ± 0.0006	
	PNN	0.8012 ± 0.0007	0.8073 ± 0.0007	0.8012 ± 0.0007	0.8008 ± 0.0009	
	LMkNN	0.7802 ± 0.0008	0.7867 ± 0.0008	0.7802 ± 0.0008	0.7798 ± 0.0008	
	LMPNN	0.7802 ± 0.0008	0.7867 ± 0.0008	0.7802 ± 0.0008	0.7798 ± 0.0008	
	PAPNN	0.8043 ± 0.0010	0.8093 ± 0.0010	0.8043 ± 0.0010	0.8038 ± 0.0008	
	FkNN	0.7839 ± 0.0010	0.7890 ± 0.0011	0.7839 ± 0.0010	0.7835 ± 0.0008	
	EigenClass	0.7978 ± 0.0007	0.8058 ± 0.0009	0.7978 ± 0.0007	0.79689 ± 0.0004	
	GMDkNN	0.7799 ± 0.0006	0.7850 ± 0.0006	0.7799 ± 0.0006	0.7796 ± 0.0008	
	BMFkNN	0.7091 ± 0.0016	0.7138 ± 0.0015	0.7091 ± 0.0016	0.7084 ± 0.0019	
	FPFS-kNN	0.8023 ± 0.0009	0.8085 ± 0.0009	0.8023 ± 0.0009	0.8018 ± 0.0011	
	SVM	0.7927 ± 0.0021	0.8033 ± 0.0020	0.7927 ± 0.0021	0.7919 ± 0.0021	
	RF	0.7975 ± 0.0002	0.8032 ± 0.0003	0.7975 ± 0.0002	0.7971 ± 0.0002	
Note:

The best classification results for each data set are shown in bold.

Table 3 The settings of the compared algorithms.

Methods	Settings	
kNN	k=1,2,...,20	
WkNN	k=1,2,...,20	
PNN	k=1,2,...,20	
LMkNN	k=1,2,...,20	
LMPNN	k=1,2,...,20	
PAPNN	k=1,2,...,20	
FkNN	k=1,2,...,20	
EigenClass	k=1,2,...,20	
GMDkNN	k=1,2,...,20,p=−2	
BMFkNN	k=1,2,...,20,p=0.5,q=3,m=1.5	
FPFS-kNN	k=1,2,...,20,Pearson	
SVM	None	
RF	None	

Accuracy, precision, recall, and F1-score and their standard deviations (SD) (Memis, Enginoǧlu & Erkan, 2022a) are used to evaluate the algorithms. Note that the best classification results for each data set are shown in bold in Table 2. It can be observed that the proposed PAPNN method achieves better performance comparing with the other twelve classifiers on the whole.

To facilitate the interpretation of the results in Table 2, ranking numbers of the best results and a pairwise comparison of the ranking results are shown in Tables 4 and 5, respectively. Note that the best ranking results are shown in bold in Table 4. It can be seen from Table 4 that PAPNN outperforms the other algorithms for 19 datasets. Besides, it is clear from Table 5 that PAPNN outperforms the other algorithms for at least 12 datasets in every metric in the pairwise comparisons. The results above illustrate that the proposed PAPNN method is superior to other methods on the whole.

Table 4 Ranking number of the best results for all algorithms compared among each other.

Methods	Accuracy	Precision	Recall	F1-Score	Total rank	
kNN	0	0	0	0	0	
WkNN	0	0	0	0	0	
PNN	0	0	0	0	0	
LMkNN	0	0	0	0	0	
LMPNN	0	0	0	1	1	
PAPNN	11	11	11	11	44	
FkNN	0	0	0	0	0	
EigenClass	0	0	0	0	0	
GMDkNN	1	1	1	1	1	
BMFkNN	0	0	0	0	0	
FPFS-kNN	0	1	0	0	1	
SVM	1	0	1	0	0	
RF	6	6	6	6	24	
Note:

The best ranking results are shown in bold.

Table 5 Ranking number of the best results for two algorithms compared vs. each other.

Methods	Accuracy	Precision	Recall	F1-score	
PAPNN vs. kNN	19	19	19	19	
PAPNN vs. WkNN	19	17	19	18	
PAPNN vs. PNN	19	19	19	19	
PAPNN vs. LMkNN	19	17	19	18	
PAPNN vs. LMPNN	19	17	19	18	
PAPNN vs. FkNN	19	17	19	18	
PAPNN vs. EigenClass	17	16	17	18	
PAPNN vs. GMDkNN	17	16	18	17	
PAPNN vs. BMFkNN	18	17	18	18	
PAPNN vs. FPFS-kNN	18	17	18	18	
PAPNN vs. SVM	16	17	16	17	
PAPNN vs. RF	13	12	13	12	

In order to enhance persuasiveness, a nonparametric statistical test called Friedman test (Demsar, 2006; Garcia & Herrera, 2008; Derrac et al., 2011) is carried out to compare the performance of classifiers. In the Friedman test, the best classifier is ranked as 1, the second best rank 2, and so on. In the case of ties, average ranks are assigned. Let Rmi be the rank of the mth of n methods on the ith of t data sets. The average rank of mth method is Rm=1t∑i=1tRim. Under the null hypothesis of the Friedman test, all competing methods nearly have similar classification performance and so the ranks Rm should be equal. The Friedman statistics is defined as follows:

(11) χF2=12tn(n+1)[∑mRm2−n(n+1)24].

When t>10 and n>5, Friedman statistics is distributed according to χF2 with n−1 degrees of freedom.

The average ranks of thirteen methods are listed in Table 6. From Table 6, It can be observed that the proposed PAPNN method achieves the best ranking. According to Eq. (11), the accuracy, precision, recall, and F1-score values of the Friedman test statistic are χF2=72.954, χF2=73.070, χF2=72.954, χF2=76.120, respectively. With 12(k−1) degrees of freedom and the critical value for the Friedman test given for with k=13 is 21.026 at a significance level of α=0.05. It can be concluded that the accuracy ( 72.954>21.026), precision ( 73.070>21.026), recall ( 72.954>21.026), and F1-score ( 76.120>21.026) values of the studied methods are significantly different. Now that the null hypothesis is rejected, a post-hoc test can be proceeded. The Nemenyi test (Demsar, 2006; Memis, Enginoǧlu & Erkan, 2022a) is used to compare all classifiers with each other.

Table 6 The average ranks of thirteen methods using Friedman test on real-world data sets.

Methods	Accuracy	Precision	Recall	F1-score	
kNN	9.31	9.5	9.36	9.42	
WkNN	8.47	8.52	8.47	8.18	
PNN	6.44	6.94	6.44	6.78	
LMkNN	5.78	5.36	5.78	5.57	
LMPNN	5.78	5.36	5.78	5.52	
PAPNN	1.84	2.42	1.84	2.26	
FkNN	8.55	8.26	8.55	8.21	
EigenClass	9.26	9.39	9.26	9.73	
GMDkNN	6.05	5.68	6.05	5.63	
BMFkNN	8.68	8.10	8.68	8.15	
FPFS-kNN	8.78	8.84	8.78	8.60	
SVM	7.34	8.31	7.39	8.84	
RF	4.55	4.21	4.55	4.05	

The critical value in the experiments with k=13 and α=0.05 is CD0.05=4.186. As a result, the accuracy, precision, recall, and F1-score of the proposed PAPNN method is significantly different from kNN, WkNN, PNN, SVM, FPFS-kNN, FkNN, and BMFkNN methods, while it is not significantly different from RF, LMkNN, LMPNN, and GMDkNN method. Figure 1 presents the critical diagrams generated by the Nemenyi post-hoc test for the four performance measures.

Figure 1 The critical diagrams for the four measures ((A) accuracy, (B) precision, (C) recall, (D) F1-score): the results from the Nemenyi post hoc test at 0.05 significance level and average rank scores from Friedman test.

Experiments on high dimensional data sets

In this section, in order to further verify the performance of the proposed PAPNN, the experiments are conducted on three high dimensional data sets in terms of accuracy, precision, recall, and F1-score comparing to other kNN-based methods. The detailed information of the datasets can be found in the last three rows of Table 1. The values of k are varied from 1 to 20 with a step of 1. The comparative results are listed in Table 7. Note that the best classification result of thirteen methods on each high data set is highlighted in bold-face. It can be seen from Table 7 that the proposed PAPNN method achieves better classification performance than other nine methods on the whole.

Table 7 Comparative results for three high dimensional datasets.

Data	Methods	Accuracy ± SD	Precision ± SD	Recall ± SD	F1-score ± SD	
AndrogenReceptor	kNN	0.8948 ± 1.45×10−5	0.8825 ± 2.81×10−5	0.8948 ± 1.45×10−5	0.8667 ± 0.0013	
	WkNN	0.9051 ± 1.02×10−5	0.8979 ± 2.26×10−5	0.9051 ± 1.02×10−5	0.8827 ± 0.0009	
	PNN	0.9044 ± 1.14×10−5	0.8931 ± 1.20×10−5	0.9044 ± 1.14×10−5	0.8827 ± 0.0009	
	LMkNN	0.8816 ± 1.21×10−5	0.88203 ± 7.25×10−6	0.8816 ± 1.21×10−5	0.8817 ± 0.0001	
	LMPNN	0.8846 ± 2.31×10−5	0.8821 ± 8.96×10−6	0.8846 ± 2.31×10−5	0.8832 ± 8.60×10−5	
	PAPNN	0.9132 ± 1.50×10−6	0.9122 ± 2.50×10−6	0.9132 ± 1.50×10−6	0.9121 ± 6.53×10−5	
	FkNN	0.8964 ± 9.36×10−6	0.8856 ± 1.51×10−5	0.8964 ± 9.36×10−6	0.8699 ± 0.0010	
	GMDkNN	0.8950 ± 1.32×10−5	0.8847 ± 1.05×10−5	0.8950 ± 1.32×10−5	0.8885 ± 2.70×10−5	
	BMFkNN	0.90737 ± 1.22×10−5	0.8951 ± 2.03×10−5	0.9073 ± 1.22×10−5	0.8926 ± 0.0003	
	FPFS-kNN	0.8970 ± 4.42×10−6	0.8849 ± 8.60×10−6	0.8970 ± 4.42×10−6	0.8700 ± 0.0005	
Semeion	kNN	0.9527 ± 6.55×10−5	0.9542 ± 4.25×10−5	0.9527 ± 6.55×10−5	0.9453 ± 0.0028	
	WkNN	0.9463 ± 1.01×10−5	0.9448 ± 9.24×10−6	0.9463 ± 1.01×10−5	0.9405 ± 0.0003	
	PNN	0.9485 ± 5.36×10−6	0.9483 ± 2.27×10−6	0.9485 ± 5.36×10−6	0.9427 ± 0.0003	
	LMkNN	0.9604 ± 6.02×10−5	0.96156 ± 4.47×10−5	0.9604 ± 6.02×10−5	0.9557 ± 0.0024	
	LMPNN	0.9672 ± 8.32×10−6	0.9674 ± 8.33×10−6	0.9672 ± 8.32×10−6	0.9645 ± 0.0003	
	PAPNN	0.9670 ± 1.73×10−6	0.9676 ± 1.41×10−6	0.9670 ± 1.73×10−6	0.9670 ± 6.71×10−5	
	FkNN	0.9533 ± 6.09×10−5	0.9549 ± 4.02×10−5	0.9533 ± 6.09×10−5	0.9462 ± 0.0026	
	GMDkNN	0.9544 ± 2.34×10−6	0.9538 ± 5.98×10−6	0.9544 ± 2.34×10−6	0.9503 ± 3.43×10−5	
	BMFkNN	0.94642 ± 1.72×10−5	0.9463 ± 1.12×10−5	0.9464 ± 1.72×10−5	0.9399 ± 0.0008	
	FPFS-kNN	0.9621 ± 7.74×10−6	0.9610 ± 9.89×10−6	0.9621 ± 7.74×10−6	0.9592 ± 0.0001	
CNAE-9	kNN	0.6279 ± 0.0060	0.7717 ± 0.0004	0.6279 ± 0.0060	0.6337 ± 0.0061	
	WkNN	0.7406 ± 6.85×10−5	0.7827 ± 5.30×10−5	0.7406 ± 6.85×10−5	0.7402 ± 6.78×10−5	
	PNN	0.7672 ± 0.0003	0.8178 ± 0.0003	0.7672 ± 0.0003	0.7660 ± 0.0003	
	LMkNN	0.7977 ± 0.0003	0.82226 ± 0.0003	0.7977 ± 0.0003	0.7995 ± 0.0003	
	LMPNN	0.8022 ± 0.0004	0.8191 ± 0.0004	0.8022 ± 0.0004	0.8031 ± 0.0004	
	PAPNN	0.8122 ± 8.54×10−6	0.8206 ± 7.67×10−6	0.8122 ± 8.54×10−6	0.8149 ± 8.10×10−6	
	FkNN	0.6554 ± 0.0034	0.7882 ± 0.0002	0.6554 ± 0.0034	0.6628 ± 0.0034	
	GMDkNN	0.8084 ± 0.0008	0.8212 ± 0.0003	0.8084 ± 0.0008	0.8058 ± 0.0008	
	BMFkNN	0.72875 ± 0.0002	0.7512 ± 0.0001	0.7287 ± 0.0002	0.7245 ± 0.0002	
	FPFS-kNN	0.7967 ± 0.0003	0.81226 ± 0.0003	0.7957 ± 0.0003	0.7985 ± 0.0003	
Note:

The best classification results for each data set are shown in bold.

Running cost analysis

From the pseudo code of the PAPNN algorithm, the computational complexity is O( Mn(n−1)) in terms of big O notation. Here, n and M are the number of the training samples and of their attributes, respectively. PAPNN has higher complexity compared to the other aforesaid algorithms. Therefore, the PAPNN algorithm requires more time to complete its execution compared to other algorithms. The mean processing time data of all aforsaid algorithms on 19 real datasets at twenty runs are listed in Tables 8–10. From Tables 8–10, it can be observed that PAPNN seems to operate significantly slower compared to other algorithms in the case of a relatively large number of sampling points and relatively high dimensionality. Despite this issue, PAPNN’s running time remains under 5 s for thirteen out of the 19 datasets.

Table 8 Average processing time for the real datasets (in seconds).

Methods	Vehicle	Balance	Blood	Bupa	Ionosphere	Pima-Indians	Parkinsons	
kNN	0.0079	0.0062	0.0079	0.0042	0.0059	0.0076	0.0029	
WkNN	0.0089	0.0069	0.0069	0.0038	0.0049	0.0079	0.0036	
PNN	0.3248	0.0747	0.0868	0.0558	0.2461	0.1416	0.0927	
LMkNN	0.3432	0.0754	0.0852	0.0560	0.2579	0.1477	0.1062	
LMPNN	0.3850	0.0811	0.0907	0.0616	0.2846	0.1552	0.1313	
PAPNN	6.1658	1.3197	2.6635	0.6329	3.3422	3.8297	0.7803	
FkNN	0.0128	0.0089	0.0099	0.0059	0.0059	0.0109	0.0049	
EigenClass	2.3763	0.6803	0.6891	0.3573	1.3862	0.8976	0.4995	
GMDkNN	0.3936	0.0903	0.1080	0.0657	0.2970	0.1595	0.1303	
BMFkNN	0.0416	0.0149	0.0169	0.0179	0.0643	0.0249	0.0538	
FPFS-kNN	1.6477	0.3550	0.4280	0.2707	1.3852	0.9241	0.6141	
SVM	0.3344	0.0579	0.0794	0.0388	0.0558	0.1136	0.0224	
RF	2.9254	1.6681	1.8746	1.5895	2.5380	3.0331	1.9394	

Table 9 Average processing time for the real datasets (in seconds).

Methods	Hill	Haberman-survival	Musk-1	Sonar	Wine	Cardiotocography	QSAR	
kNN	0.0079	0.0062	0.0079	0.0042	0.0059	0.0076	0.0029	
WkNN	0.0089	0.0069	0.0069	0.0038	0.0049	0.0079	0.0036	
PNN	0.3248	0.0747	0.0868	0.0558	0.2461	0.1416	0.0927	
LMkNN	0.3432	0.0754	0.0852	0.0560	0.2579	0.1477	0.1062	
LMPNN	0.3850	0.0811	0.0907	0.0616	0.2846	0.1552	0.1313	
PAPNN	199.6386	0.3864	35.6519	2.8254	0.2553	27.2365	46.0123	
FkNN	0.0128	0.0089	0.0099	0.0059	0.0059	0.0109	0.0049	
EigenClass	2.3763	0.6803	0.6891	0.3573	1.3862	0.8976	0.4995	
GMDkNN	0.3936	0.0903	0.1080	0.0657	0.2970	0.1595	0.1303	
BMFkNN	0.0416	0.0149	0.0169	0.0179	0.0643	0.0249	0.0538	
FPFS-kNN	1.6477	0.3550	0.4280	0.2707	1.3852	0.9241	0.6141	
SVM	0.3344	0.0579	0.0794	0.0388	0.0558	0.1136	0.0224	
RF	2.9254	1.6681	1.8746	1.5895	2.5380	3.0331	1.9394	

Table 10 Average processing time for the real datasets (in seconds).

Methods	Band	Pima	Wine (keel)	Mammographic	Steel	
kNN	0.0056	0.0079	0.0038	0.0219	0.0087	
WkNN	0.0045	0.0078	0.0040	0.0242	0.0069	
PNN	0.1497	0.1597	0.0558	1.1043	0.1116	
LMkNN	0.1568	0.1471	0.0630	1.1276	0.1120	
LMPNN	0.1730	0.1568	0.0823	1.2106	0.1141	
PAPNN	2.0690	4.2120	0.2546	34.710	3.2976	
FkNN	0.0109	0.0170	0.0089	0.0388	0.0159	
EigenClass	1.2020	1.4945	0.4132	8.1129	1.2795	
GMDkNN	0.2523	0.2303	0.1296	1.7251	0.1576	
BMFkNN	0.0390	0.0239	0.0358	0.0627	0.0197	
FPFS-kNN	0.9335	0.9516	0.3944	9.0507	0.9944	
SVM	0.0662	0.1055	0.0187	1.5800	0.1000	
RF	2.1755	2.7420	1.3017	9.3959	2.3380	

Conclusion

In this article, a pre-averaged based pseudo nearest neighbor classifier (PAPNN) is proposed to improve the classification performance in the case of small-size training samples with existing outliers. The PAPNN rule, which is based on the average idea, can further reduce the negative impact of existing outliers to some degree. Classification performance of PAPNN is evaluated through an effective comparison with kNN, WkNN, PNN, LMkNN, LMPNN, FkNN, GMDkNN, BMFkNN, FPFS-kNN, EigenClass, SVM, and RF. The algorithms are trained and tested for 10 runs using 10-fold cross-validation over the nineteen real numerical data sets and three high dimensional data sets. The results are then evaluated using several well-known measures, such as accuracy, precision, recall, and F1-score. Statistical analysis and computational complexity are carried out to further check the accuracy and processing time of the algorithm by comparing with the other algorithms.

The experimental results and statistical analysis show that PAPNN has the best classification performance on the whole. This issue is the significant advantage of PAPNN. On the other hand, PAPNN’s disadvantage is that the processing time will become longer with the number of training set points increasing. This disadvantage may limit PAPNN to perform large-scale data classification on personal computers. To deal with this disadvantage, distributed computing or the hardware accelerator, such as graphics processing unit (GPU), can be employed to accelerate the classification process.

In general, the data at the class boundary is uncertain to some extent. The fuzzy algorithm can deal with this uncertainty to some degree, thus improving the accuracy and robustness of classification. Therefore, the future research should be focused on combining PAPNN with the fuzzy algorithm to improve the classification effect.

Supplemental Information

Supplemental Information 1 The specific implementation code of the PAPNN method proposed in this article.

The code mainly consists of the following parts: (1) Initialize and read the dataset into memory; (2) Normalize the data; (3) Implement the PAPNN algorithm program; (4) Implement the classification result statistics function.

Supplemental Information 2 Datasets involved in the experiments of this article.

Supplemental Information 3 Dataset path file.

This file is required by the algorithm program proposed in this paper. Its specific content is the path on the disk where the experimental datasets are stored.

Additional Information and Declarations

Competing Interests

Author Contributions

Data Availability

The authors declare that they have no competing interests.

Dapeng Li conceived and designed the experiments, performed the experiments, analyzed the data, performed the computation work, prepared figures and/or tables, authored or reviewed drafts of the article, and approved the final draft.

The following information was supplied regarding data availability:

Data and code are available in the Supplemental Files.

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
