# Peer review of "A pre-averaged pseudo nearest neighbor classifier"

_PeerJ Computer Science, doi:10.7717/peerj-cs.2247_

## Round 0.1 · original submission · Major Revisions

I have received the review reports for your paper submitted to PeerJ Computer Science from the reviewers. According to the reports, I will recommend major revision to your paper. Please refer to the reviewers’ opinions to improve your paper. Please also write a revision note such that the reviewers can easily check whether their comments are fully addressed. We look forward to receiving your revised manuscript soon.

Reviewers 1 & 2 have suggested that you cite specific references. You are welcome to add it/them if you believe they are relevant. However, you are not required to include these citations, and if you do not include them, this will not influence my decision.

With best regards

**Language Note:** PeerJ staff have identified that the English language needs to be improved. When you prepare your next revision, please either (i) have a colleague who is proficient in English and familiar with the subject matter review your manuscript, or (ii) contact a professional editing service to review your manuscript. PeerJ can provide language editing services - you can contact us at [email protected] for pricing (be sure to provide your manuscript number and title). – PeerJ Staff

Reviewer 1 ·

Basic reporting

The review report has been attached.

Experimental design

The review report have been attached.

Validity of the findings

The review report have been attached.

Additional comments

The review report have been attached.

Annotated reviews are not available for download in order to protect the identity of reviewers who chose to remain anonymous.
Cite this review as
Anonymous Reviewer (2024) Peer Review #1 of "A pre-averaged pseudo nearest neighbor classifier (v0.1)". PeerJ Computer Science

Reviewer 2 ·

Basic reporting

#1- The authors should have used a grammar check editor for some grammatical errors and misprints.
#2- The introduction section should be extended by citing up-to-date studies related to machine learning before focusing kNN-based studies.
#3- End of the Introduction Section, the paragraph related to organizing of the paper should be added.
#4- The author(s) provided the figures with high-resolution. It is very reader-friendly.

Experimental design

#1- In Table 1, whether the datasets are balance or imbalance should be provided.
#2- The settings of the compared algorithms should be summarised in a table.
#3- Why are the 10-fold cross validation chosen? It should be explained.
#4- The number of runs should be indicated.
#5- The compared algorithms are not up to date. In other words, the kNN (1967) and WkNN (1976) are primary studies, however; LMkNN (2006), PNN (2009), and LMPNN (2014) are not up to date. Well-known Fuzzy kNN and the recent studies should be mentioned and at least two of recent ones should be compared. For example,
Fuzzy kNN - A fuzzy k-nearest neighbor algorithm, IEEE Trans. Syst. Man Cybern. 15 (1985) 580–585, https://doi.org/10.1109/TSMC.1985.6313426
BM-Fuzzy kNN - A new fuzzy k -nearest neighbor classifier based on the Bonferroni mean, Pattern Recogn. Lett. 140 (2020), 172–178, https://doi.org/10.1016/j.patrec.2020.10.005
EigenClass - A Precise and Stable Machine Learning Algorithm: Eigenvalue Classification (EigenClass), Neural Computing and Applications, 33(10), 5381–5392 (2021). https://doi.org/10.1007/s00521-020-05343-2
FPFS-AC - A new classification method using soft decision-making based on an aggregation operator of fuzzy parameterized fuzzy soft matrices. Turkish Journal of Electrical Engineering and Computer Sciences, 30(3) (2022), 871-890.
PFS-kNN - Picture Fuzzy Soft Matrices and Application of Their Distance Measures to Supervised Learning: Picture Fuzzy Soft k-Nearest Neighbor (PFS-kNN), Electronics, 12(19) (2023), 4129; https://doi.org/10.3390/electronics12194129
#6- For the comparison, using standard deviation and error rate are not adequate. Accuracy, Precision, Recall, F1-Score should be utilized for performance metrics. Here, the metrics different for binary classification and more.
#7- The comparison results should be analysed using statistical techniques in detail. The Friedman test is utilized for determining average ranks. In addition, Nemenyi tests can be employed for Nemenyi diagrams. For more details, the followings can be considered:
(Cited in the manuscript) Statistical comparisons of classifiers over multiple data sets, J. Mach. Learn. Res. 7 (2006) 1–30. https://www.jmlr.org/papers/volume7/demsar06a/demsar06a.pdf.
A classification method in machine learning based on soft decision-making via fuzzy parameterized fuzzy soft matrices, Soft Comput. 26 (2022) 1165–1180, https://doi.org/10.1007/s00500-021-06553-z

Validity of the findings

#1- The conclusion looks insufficient. It should be completely re-written.
#2- After the obtaining new simulation results, conclusion section should be highlighted by mentioning advantages, disadvantages, and limitations of the study.
#3- The future works should be provided in detail.

Additional comments

Dear Author(s),

I have also attached the review report. Please consider it.

Best regards,

Annotated reviews are not available for download in order to protect the identity of reviewers who chose to remain anonymous.
Cite this review as
Anonymous Reviewer (2024) Peer Review #2 of "A pre-averaged pseudo nearest neighbor classifier (v0.1)". PeerJ Computer Science

Reviewer 3 ·

Basic reporting

In this papers the author presents a pre-averaged pseudo nearest neighbor classifier, which is effective for classification task in the case of small-size samples with existing outliers. The paper is well written but the novelty is low with respect to other publications of the authors, such as:

Zheng Chai, Yanying Li, Aili Wang, Chen Li, Baoshuang Zhang and Huanhuan Gong. An Efficient Pseudo Nearest Neighbor Classifier. IAENG International Journal of Computer Science, 48:4, IJCS_48_4_26

The proposal is aimed at improving the classification on small sets and with outliers, as the authors comment that this is especially important in this type of approaches. In my opinion, in neighborhood-based approaches, regardless of the size of the problem, the most important issue is to perform a quality selection of the k nearest neighbors to avoid noise problems and not to consider examples that are really very different from the example we want to classify. In fact, nowadays, it is usual to have huge amounts of data that we have to process. The problem is that the computation of pre-measured vectors would be very expensive in these cases. In the paper, this issue is not explicitly explained, and it may be worthwhile to insist a little on this point.

The experiment should be improved:
- The author should make a study of how its proposal behaves in High Dimensional Problems, including a scalability analysis of the proposal.
- The author should include in the comparative study its proposals published in other papers to highlight the advantages of his new proposal.
- The author should add in its study current proposals based on other approaches than neighborhood based (tree ensemble, SVM, etc) to analyze the predictive capability of your proposal. It is clear that your proposal improves the results of previous neighborhood-based methods but you should highlight their predictive capability with respect to other approaches.
- The author applies a Friedman test but then it is necessary to apply a nonparametric multiple comparison test to compare the method with the best ranking with the rest of the methods analyzed.

Experimental design

The experiment should be improved:
- The author should make a study of how its proposal behaves in High Dimensional Problems, including a scalability analysis of the proposal.
- The author should include in the comparative study its proposals published in other papers to highlight the advantages of his new proposal.
- The author should add in its study current proposals based on other approaches than neighborhood based (tree ensemble, SVM, etc) to analyze the predictive capability of your proposal. It is clear that your proposal improves the results of previous neighborhood-based methods but you should highlight their predictive capability with respect to other approaches.
- The author applies a Friedman test but then it is necessary to apply a nonparametric multiple comparison test to compare the method with the best ranking with the rest of the methods analyzed.

Validity of the findings

As I mentioned in the previous point, the experiments must be improved.

Cite this review as
Anonymous Reviewer (2024) Peer Review #3 of "A pre-averaged pseudo nearest neighbor classifier (v0.1)". PeerJ Computer Science

---

## Round 0.2 · Major Revisions

I have received the review reports for your paper submitted to PeerJ Computer Science from the reviewers. According to the reports, I will recommend major revision to your paper. Please refer to the PDF from Reviewer 2. Please provide a rebuttal letter such that the reviewers can easily check whether their comments are fully addressed. We look forward to receiving your revised manuscript soon.

Reviewer 1 ·

Basic reporting

I have reviewed the revised paper. It is observed that the author has made the requested changes and improvements. My evaluation is that the paper is eligible for publication in this journal.

Experimental design

I have reviewed the revised paper. It is observed that the author has made the requested changes and improvements. My evaluation is that the paper is eligible for publication in this journal.

Validity of the findings

I have reviewed the revised paper. It is observed that the author has made the requested changes and improvements. My evaluation is that the paper is eligible for publication in this journal.

Additional comments

I have reviewed the revised paper. It is observed that the author has made the requested changes and improvements. My evaluation is that the paper is eligible for publication in this journal.

Cite this review as
Anonymous Reviewer (2024) Peer Review #1 of "A pre-averaged pseudo nearest neighbor classifier (v0.2)". PeerJ Computer Science

Reviewer 2 ·

Basic reporting

I have attached the review report. Please consider it.

Experimental design

I have attached the review report. Please consider it.

Validity of the findings

I have attached the review report. Please consider it.

Additional comments

I have attached the review report. Please consider it.

Annotated reviews are not available for download in order to protect the identity of reviewers who chose to remain anonymous.
Cite this review as
Anonymous Reviewer (2024) Peer Review #2 of "A pre-averaged pseudo nearest neighbor classifier (v0.2)". PeerJ Computer Science

---

## Round 0.3 · accepted · Accept

The authors have fully addressed the comments of the reviewers. I am happy to make a decision of acceptance to the paper.

Reviewer 2 ·

Basic reporting

In the revised manuscript, all the suggestions have been adopted and the major corrections have been made. Therefore, my evaluation is that the revised version of this manuscript can be published in this journal.

Experimental design

In the revised manuscript, all the suggestions have been adopted and the major corrections have been made. Therefore, my evaluation is that the revised version of this manuscript can be published in this journal.

Validity of the findings

In the revised manuscript, all the suggestions have been adopted and the major corrections have been made. Therefore, my evaluation is that the revised version of this manuscript can be published in this journal.

Additional comments

Dear Authors,

In the revised manuscript, all the suggestions have been adopted and the major corrections have been made. Therefore, my evaluation is that the revised version of this manuscript can be published in this journal.

Best regards,

Cite this review as
Anonymous Reviewer (2024) Peer Review #2 of "A pre-averaged pseudo nearest neighbor classifier (v0.3)". PeerJ Computer Science